# Stragglers Can Contribute More: Uncertainty-Aware Distillation for Asynchronous Federated Learning

## Abstract

Asynchronous federated learning (FL) has recently gained attention for its enhanced efficiency and scalability, enabling local clients to send model updates to the server at their own pace without waiting for slower participants. However, such a design encounters significant challenges, such as the risk of outdated updates from straggler clients degrading the overall model performance and the potential bias introduced by faster clients dominating the learning process, especially under heterogeneous data distributions. Existing methods typically address only one of these issues, creating a conflict where mitigating the impact of outdated updates can exacerbate the bias created by faster clients, and vice versa. To address these challenges, we propose FedEcho, a novel framework that incorporates uncertainty-aware distillation to enhance the asynchronous FL performances under large asynchronous delays and data heterogeneity. Specifically, uncertainty-aware distillation enables the server to assess the reliability of predictions made by straggler clients, dynamically adjusting the influence of these predictions based on their estimated uncertainty. By prioritizing more certain predictions while still leveraging the diverse information from all clients, FedEcho effectively mitigates the negative impacts of outdated updates and data heterogeneity. Through extensive experiments, we demonstrate that FedEcho consistently outperforms existing asynchronous federated learning baselines, achieving robust performance without requiring access to private client data.

## 1 Introduction

Federated learning (FL) (McMahan et al., 2017) has emerged as a promising paradigm of collaboratively training machine learning models across edge clients without sharing private data. In the standard synchronous FL setting, the central server waits for all assigned clients to complete their local training before aggregating the local updates to the global model (McMahan et al., 2017; Karimireddy et al., 2020; Wang et al., 2022; 2020b; Wei et al., 2020). However, this synchronous aggregation scheme suffers from poor scalability in heterogeneous environments due to system discrepancies: faster clients remain idle while waiting for straggler clients, leading to significant inefficiency in training (Xie et al., 2019; Nguyen et al., 2022; Yang et al., 2022).

Asynchronous FL alleviates the inefficiency by allowing clients to upload updates at their own pace, enabling the faster clients and the server to process global updates without waiting for all participant clients (Xie et al., 2019; Nguyen et al., 2022; Toghani & Uribe, 2022; Wang et al., 2024c;d; Liu et al., 2024). Although asynchronous methods improve training efficiency, they do not come without costs. Updates from straggler clients usually suffer from asynchronous delay, as their local models are trained on an outdated global model rather than the current one. Consequently, incorporating such outdated updates into the global model would impede the overall convergence, especially when encountering *large asynchronous delays*. Meanwhile, faster clients usually contribute more frequent updates due to smaller delays, and thus cause the global model to be biased towards their data distributions. This would deteriorate the overall performance in the presence of *heterogeneous client data distributions*.

Despite several prior works attempting to enhance the performance of asynchronous FL, the two issues appear to be in conflict, with existing research primarily addressing only one of them. Specifically, one line of work aims to balance the client contributions in the global update by using the historical model updates or applying gradient correction (Wang et al., 2024c; Zang et al., 2024). However, such methods directly merge the outdated parameter updates from straggler clients into the global model, which distorts the training and leads to degraded performance under large delays[1]. Another line of work aims to reduce the effect of outdated updates by employing momentum-based training design (Yu et al., 2024; Wang et al., 2024d), thus reducing the impact brought by straggler clients. However, since the momentum terms are even more dominated by frequent updates from faster clients, the model updates of stragglers are gradually diminished, resulting in their updates being largely neglected and finally leading to convergence slowdown under severe data heterogeneity. This motivates us to explore the following question:

> *Can we solve the two above-mentioned issues simultaneously by developing a new asynchronous FL approach that works under both* **large asynchronous delays** *and* **client data heterogeneity***?*

Specifically, this necessitates a more nuanced approach to managing outdated client updates from straggler clients: on one hand, we aim to avoid directly merging or averaging these outdated updates into the global model to prevent performance degradation; on the other hand, we seek to retain the useful information from the outdated updates, as they hold valuable insights regarding the unique data distributions of the clients.

Based on this, we propose **FedEcho**, a novel uncertainty-aware distillation framework for asynchronous FL. Building upon knowledge distillation, FedEcho operates at the prediction logits level to update model parameters, thereby avoiding direct parameter contamination from straggler updates. Specifically, FedEcho aggregates client predictions into ensemble teacher logits and performs server-side distillation. To further make sure that FedEcho could retain useful information from the outdated updates, we design *uncertainty-aware distillation*: predictions obtained from straggler clients (who trained on outdated models), may still be noisy and unreliable. This motivates the need for uncertainty-aware distillation, where the server dynamically adjusts its confidence in the teacher's predictions based on prediction uncertainty. This makes FedEcho a robust framework that effectively leverages the insights from straggler updates while minimizing the risks associated with outdated information. By integrating uncertainty-aware distillation, FedEcho ensures that the global model benefits from the diverse data distributions of all clients, fostering more accurate and reliable learning outcomes in the presence of both large asynchronous delays and heterogeneous client data.

Our main contributions are summarized as follows:

- We propose FedEcho, a novel asynchronous FL framework that incorporates uncertainty-aware distillation to effectively address the challenges posed by large asynchronous delays and heterogeneous client data distributions. FedEcho builds upon the principles of knowledge distillation, enabling the dynamic adjustment of distillation loss based on uncertainty, while ensuring balanced contributions from all clients.

- We provide theoretical convergence analysis of the proposed FedEcho and with a convergence rate of $\mathcal{O}(\frac{1}{\sqrt{TM}})$ w.r.t. the number of global communication rounds $T$, and the number of accumulated gradients $M$. This rate is consistent with that of asynchronous FL methods, ensuring that the involvement of uncertainty-aware distillation does not compromise the convergence.

- We conduct extensive experiments under varying degrees of data heterogeneity and asynchronous delay. The results demonstrate that FedEcho consistently outperforms existing asynchronous FL baselines under various experimental settings.

---

[1]While Wang et al. (2024c) mentioned alleviating the joint effect of asynchronous delay and client data heterogeneity, experimental results suggest that their effectiveness deteriorates under large asynchronous delays.

## 2 RELATED WORKS

**Asynchronous FL** Asynchronous optimizations and their various adaptations, such as Hog-wild (Niu et al., 2011) and asynchronous SGD (Mania et al., 2017; Nguyen et al., 2018; Stich et al., 2021; Leblond et al., 2018; Glasgow & Wootters, 2022), have been widely discussed due to their advantage in computational efficiency and scalability. Recently, asynchronous update algorithms in FL, including FedAsync (Xie et al., 2019) and FedBuff (Nguyen et al., 2022), have also garnered significant attention, as they provide a more flexible and efficient solution for adapting to client systematic heterogeneity. Based on FedBuff (Nguyen et al., 2022), recent works have focused on various aspects of asynchronous FL, including theoretical analysis (Toghani & Uribe, 2022), communication efficiency (Ortega & Jafarkhani, 2023), and data heterogeneity (Wang et al., 2024c;b). Moreover, recent studies have investigated the integration of advanced optimization techniques such as momentum acceleration (Yu et al., 2024; Zang et al., 2024) and adaptive optimization (Wang et al., 2024d) within asynchronous FL frameworks.

**Federated Distillation** Federated distillation has been widely studied for knowledge transfer in FL, addressing data heterogeneity and model heterogeneity issues. Early works such as FedKD Jeong et al. (2018) transmit the mean of client logits to the server, which are then used as teacher signals for local distillation regularization. Several studies focus on enhancing robustness in the face of data heterogeneity. For example, FedDF (Lin et al., 2020) ensembles soft labels from unlabeled data, Itahara et al. (2021) proposes entropy reduction aggregation, and Zhang et al. (2022) proposes a data-free knowledge distillation method to fine-tune the global model on the server. Another line of work is model heterogeneity, e.g., FedMD (Li & Wang, 2019) aligns client logits on a public dataset, FedGKT (He et al., 2020) transfers the knowledge from small client models to a large server-side model, and FedType (Wang et al., 2024a) deploys lightweight proxy models on clients to exchange knowledge with private large models, thereby eliminating the need for private data. There are also some prototype-based approaches (Tan et al., 2022; Zhang et al., 2024), multilevel distillation methods (Khan et al., 2024; Hao et al., 2023; Qi et al.) for FL.

## 3 THE PROPOSED METHOD: FEDECHO

### 3.1 PRELIMINARIES

Generally, in FL frameworks, we aim to minimize the following objective through $N$ local clients:

$$\min_{\boldsymbol{x} \in \mathbb{R}^d} f(\boldsymbol{x}) := \frac{1}{N} \sum_{i=1}^{N} F_i(\boldsymbol{x}) = \frac{1}{N} \sum_{i=1}^{N} \mathbb{E}_{\xi \sim \mathcal{D}_i}[F_i(\boldsymbol{x}; \xi_i)], \tag{1}$$

where $\boldsymbol{x}$ represents the model parameters with $d$ dimensions, $F_i(\boldsymbol{x}) = \mathbb{E}_{\xi \sim \mathcal{D}_i}[F_i(\boldsymbol{x}, \xi_i)]$ represents the local loss function corresponding to client $i$, and $\mathcal{D}_i$ denotes the local data distribution. Based on FedAvg (McMahan et al., 2017) and its variants, synchronous FL algorithms solve Eq. (1) by allowing each participating client to perform local updates, and the server receives the local updates from assigned clients and aggregates them to update the global model.

**Asynchronous updates with buffer.** To alleviate this inefficiency and enhance scalability, asynchronous federated learning has been introduced as a solution for optimizing the objective in Eq. (1). In asynchronous FL, clients conduct local training independently and upload their updates to the server upon completion. Building upon the first few asynchronous works such as FedAsync (Xie et al., 2019), FedBuff (Nguyen et al., 2022) FADAS (Wang et al., 2024d) and CA$^2$FL (Wang et al., 2024c), they utilize a buffer-based mechanism to improve global updates. Such asynchronous methods maintain a fixed number of concurrency $M_c$ (the server typically ensures that $M_c$ clients are actively performing local training simultaneously, with $M_c$ usually determined by the server). Then at global round $t$, when client $i$ finishes $K$ steps of local training, it sends the update $\boldsymbol{\Delta}_t^i = \boldsymbol{x}_{t-\tau,K}^i - \boldsymbol{x}_{t-\tau}$ to the server, where $t - \tau$ indicates the global round when local training began. The server accumulates the local updates to $\boldsymbol{\Delta}_t$, and dispatches the latest global model to a randomly selected idle client for local training. Once $M$ updates are collected, the server updates the global model. Clients who are performing local training still continue with their current local models, which would be unaffected by global updates. Through this asynchronous design, the FL

training system maintains a fixed number of concurrency by assigning a new idle client to training whenever one finishes.

### 3.2 FEDECHO: UNCERTAINTY-ARARE DISTILLATION FOR ASYNCHRONOUS FL

To address the challenges related to stale update and achieve better convergence, we propose **FedEcho**, a novel uncertainty-aware distillation methods for asynchronous FL. Our intuition is that local models can provide coarse guidance on how the global model should adapt to diverse local data distributions. We summarize the proposed FedEcho in Algorithm 1. FedEcho adopts a standard local training procedure similar to Nguyen et al. (2022); Wang et al. (2024c), where each client uploads its model update upon completing local training. FedEcho also maintains the concept of server concurrency $M_c$ and buffer size $M$ for flexible control of the number of active clients and the frequency of global model update.

**Uncertainty-aware distillation.** At the server, FedEcho employs a novel distillation process to enable the global model to learn from local models. Upon receiving an update $\mathbf{\Delta}_t^i$ from client $i$, the server temporarily constructs a client-specific model $\boldsymbol{x}_t^i$, performs inference on an unlabeled dataset $\mathcal{U}$, and stores the resulting logits $\boldsymbol{y}^i$ for future distillation. The temporary model is discarded immediately after inference. Once the server has aggregated $M$ local updates to obtain a new global model $\widehat{\boldsymbol{x}}_{t+1}$, it performs knowledge distillation using the stored logits. Specifically, for each batch $B$ from the unlabeled dataset $\mathcal{U}$, the server aggregates the logits from all teacher models, i.e., the most recent stored logits from each client (if available), and takes their average prediction $\boldsymbol{y}_t$ as the distillation target (Line 14).

For the distillation process, the available teacher (client) models, which represented by their stored output logits on the server, are simultaneously used for the knowledge distillation:

$$\widehat{\boldsymbol{x}}_{t+1} \leftarrow \widehat{\boldsymbol{x}}_{t+1} - \eta_d \cdot \mathrm{Clip}(\nabla f_d(\boldsymbol{y}_t, \widehat{\boldsymbol{y}}_t), \nu), \quad \widehat{\boldsymbol{y}}_t = \mathrm{Logits}(\widehat{\boldsymbol{x}}_{t+1}(B)) \tag{2}$$

where $\eta_d$ is the distillation learning rate, $\nu$ is the clipping threshold. We adopt a novel *uncertainty-aware distillation loss* $f_d$ which weighting the Kullback–Leibler (KL) divergence loss and Cross-Entropy (CE) loss,

$$f_d(\boldsymbol{y}_t, \widehat{\boldsymbol{y}}_t) = \alpha \mathrm{KL}\big(\sigma(\boldsymbol{y}_t) \| \sigma(\widehat{\boldsymbol{y}}_t)\big) + (1 - \alpha)\mathrm{CE}\big(\widehat{\boldsymbol{y}}_t, \arg\max(\boldsymbol{y}_t)\big), \tag{3}$$

where $\sigma$ is the softmax function. To dynamically adjust between the richer uncertainty information which encourage by KL divergence loss and the less uncertain information providing by the CE loss, we define $\alpha$ as a function of predictive uncertainty. Given teacher logits $\boldsymbol{y}_t^u$ for sample $u$, we compute and the entropy $H^u$ and the normalized batch entropy $\widehat{H}$ for batch $B$:

$$H^u = -\sum_{c=1}^{C} p_c^u \log p_c^u, \quad \widehat{H} = \frac{1}{\log C} \frac{1}{|B|} \sum_{u \in B} H^u. \tag{4}$$

We set $\alpha$ to interpolate between $\alpha_{\min}$ and $\alpha_{\max}$ according to $\widehat{H}$, i.e., $\alpha = \widehat{H}\alpha_{\max} + (1 - \widehat{H})\alpha_{\min}$. In general, when the batch of teacher logits exhibits high entropy, the weight $\alpha$ increases and the contribution of the CE loss is reduced. This design reflects the intuition that when teacher predictions are highly uncertain, the hard labels derived from them may be unreliable, and relying more on soft labels helps the global student model avoid learning from inaccurate or noisy signals. Conversely, when the teacher logits are more confident (i.e., with low entropy $\widehat{H}$), a larger proportion of CE loss is emphasized, allowing the student model to benefit from more deterministic and trustworthy supervision.

Moreover, to prevent the imperfect guidance during the early stages of training when teacher logits may be inaccurate and fail to fully represent the local knowledge. We employ gradient clipping to constrain the magnitude of the distillation gradient as shown in Eq. (2). This ensures that the global model can still learn from coarse signals of local models, thereby enhancing resilience to the staleness and heterogeneous data distribution.

In a nutshell, the proposed FedEcho decouples the contribution of stale clients' information from the current global model update by avoiding direct aggregate auxiliary variables to the global model. Instead, it leverages knowledge distillation to incorporate information from local models. Intuitively, FedEcho captures the predictive behavior of each client: the server learns from the averaged logits

---

**Algorithm 1** FedEcho: Uncertainty-aware distillation for asynchronous federated learning

---

**Input:** local learning rate $\eta_l$, global learning rate $\eta$, distillation learning rate $\eta_d$, server concurrency $M_c$, buffer size $M$

1: Initialize model $\boldsymbol{x}_1$, initialize $\boldsymbol{\Delta}_1 = \boldsymbol{0}$, $m = 0$, server first sample a set of $\mathcal{M}_0$ with size $M_c$ of active clients to run **local SGD updates** with local learning rate $\eta_l$, logits list $\boldsymbol{y} = \boldsymbol{0}$;
2: **repeat**
3:     **if** receive client update **then**
4:         Server accumulates update from client $i$: $\boldsymbol{\Delta}_t \leftarrow \boldsymbol{\Delta}_t + \boldsymbol{\Delta}_t^i$ and set $m \leftarrow m + 1$;
5:         Server infers client $i$'s local model $\boldsymbol{x}_t^i = \boldsymbol{x}_{t-\tau_t^i} + \boldsymbol{\Delta}_t^i$ on the unlabeled set $\mathcal{U}$ and stores the logits $\boldsymbol{y}^i$ for client $i$;
6:         Sample another client $j$ from available clients;
7:         Send the current model $\boldsymbol{x}_t$ to client $j$, and run **local SGD updates** on client $j$;
8:     **end if**
9:     **if** $m = M$ **then**
10:         $\boldsymbol{\Delta}_t \leftarrow \frac{\boldsymbol{\Delta}_t}{M}$;
11:         Update global model $\widehat{\boldsymbol{x}}_{t+1} = \boldsymbol{x}_t + \eta \boldsymbol{\Delta}_t$;
12:         **for** Server distillation steps $q = 1$ to $Q$ **do**
13:             Sample an unlabeled mini-batch $B$ from unlabeled set $\mathcal{U}$;
14:             Get the teacher logits $\boldsymbol{y}_t = \frac{1}{|\mathcal{S}_u|} \sum_{i \in \mathcal{S}_u} \boldsymbol{y}^i[B]$, where $\mathcal{S}_u = \{i | y^i[B]$ is available$\}$ , and student logits $\widehat{\boldsymbol{y}}_t = \text{Logits}(\widehat{\boldsymbol{x}}_{t+1}(B))$;
15:             Update global student model via knowledge distillation $\widehat{\boldsymbol{x}}_{t+1} \leftarrow \widehat{\boldsymbol{x}}_{t+1} - \eta_d \cdot \text{Clip}(\nabla f_d(\boldsymbol{y}_t, \widehat{\boldsymbol{y}}_t), \nu)$;
16:         **end for**
17:         Update global model $\boldsymbol{x}_{t+1} \leftarrow \widehat{\boldsymbol{x}}_{t+1}$; set $m \leftarrow 0$, $\boldsymbol{\Delta}_{t+1} \leftarrow \boldsymbol{0}$, $t \leftarrow t + 1$;
18:     **end if**
19: **until** convergence

---

provided by local models and uses them as soft targets. This enables the global model to assimilate diverse client knowledge in a more stable and delay-tolerant manner. Importantly, even clients with large delays can still contribute meaningful label-distribution signals to the global model through their predictions. As a result, the global update is less dominated by recently updated or faster clients, thereby mitigating the negative effects of data heterogeneity and asynchronous delay.

**Memory overhead on the server.** While FedEcho is designed to address the impact of high latency under data heterogeneity through server-side distillation, it implicitly introduces additional memory overhead compared to standard optimization-based asynchronous FL methods (Nguyen et al., 2022; Wang et al., 2024d). Specifically, as shown in Line 5 of Algorithm 1, the server needs to temporarily construct a client-specific model $\boldsymbol{x}_t^i$ for inference. This requires the server storing the corresponding global model checkpoints $\boldsymbol{x}_{t-\tau_t^i}$ used by client $i$ at the start of its local training. However, since multiple clients may share the same initialization, the server only needs to store at most $M_c$ global model checkpoints, where $M_c$ is the concurrency number (i.e., the maximum number of concurrently active clients). In addition, the server needs to store the logits copy for each client (no matter whether actively calculating the gradient or not), yet this memory cost is modest. As FedEcho only requires hundreds to a few thousand distillation samples, the total size of stored logits is typically on the same order of magnitude as a single model copy.

**Discussion on privacy-preserving concerns.** Local data privacy is usually one of the most critical factors in federated learning. Unlike many distillation methods that rely on data from the training domain, the proposed FedEcho can achieve the desired performance by utilizing an unlabeled dataset $\mathcal{U}$ without getting the data distribution from the local clients' training data. Moreover, as shown in our experiments, FedEcho also supports the use of synthetic data generated from diffusion models for distillation, helping to alleviate practical limitations of unlabeled datasets. Furthermore, the local model update and communication process in FedEcho closely follows the standard asynchronous FL framework and does not introduce additional privacy risks. It is also compatible with privacy-preserving techniques such as differential privacy.

# 4 THEORETICAL ANALYSIS

In this section, we first study the convergence analysis under general non-convex settings for the proposed method. Moreover, we investigate the distillation gradient with one student and $N$ teacher models in Appendix B.

In this section, we delve into the convergence analysis of our proposed FedEcho algorithm. We first introduce some common assumptions required for the analysis.

**Assumption 4.1** (Smoothness). Each objective function on the $i$-th worker $F_i(\boldsymbol{x})$ is $L$-smooth, i.e., $\forall \boldsymbol{x}, \boldsymbol{y} \in \mathbb{R}^d$,

$$\|\nabla F_i(\boldsymbol{x}) - \nabla F_i(\boldsymbol{y})\| \leq L \|\boldsymbol{x} - \boldsymbol{y}\|.$$

**Assumption 4.2** (Bounded Variance). Each stochastic gradient is unbiased and has a bounded local variance, i.e., for all $\boldsymbol{x}, i \in [N]$, we have $\mathbb{E}\big[\|\nabla F_i(\boldsymbol{x}; \xi) - \nabla F_i(\boldsymbol{x})\|^2\big] \leq \sigma_l^2$, for the distillation loss $f_d$, the stochastic gradient is unbiased and has a bounded local variance as well, i.e., $\mathbb{E}\big[\|\nabla f_d(\boldsymbol{x}|\boldsymbol{x}^1, ..., \boldsymbol{x}^n; \xi) - \nabla f_d(\boldsymbol{x}|\boldsymbol{x}^1, ..., \boldsymbol{x}^n)\|^2\big] \leq \sigma_d^2$ and the loss function on each client has a global variance bound, $\frac{1}{N} \sum_{i=1}^{N} \|\nabla F_i(\boldsymbol{x}) - \nabla f(\boldsymbol{x})\|^2 \leq \sigma_g^2$.

Assumptions 4.1 and 4.2 are standard assumptions in federated non-convex optimization literature (Li et al., 2019; Yang et al., 2021; Reddi et al., 2021; Wang et al., 2022; Wang & Ji, 2023). The global variance upper bound of $\sigma_g^2$ in Assumption 4.2 measures the data heterogeneity across clients, and a global variance of $\sigma_g^2 = 0$ indicates a uniform data distribution across clients.

**Assumption 4.3** (Bounded Delay of Gradient Computation). Let $\tau_t^i$ represent the delay for global round $t$ and client $i$ which is applied in Algorithm 1. The delay $\tau_t^i$ is the difference between the current global round $t$ and the global round at which client $i$ started to compute the gradient. We assume that the maximum gradient delay (worst-case delay) is bounded, i.e., $\tau_{\max} = \max_{t \in [T], i \in [N]}\{\tau_t^i\} < \infty$. Moreover, we further define the average of the maximum delay over time $\tau_{\text{avg}} = \frac{1}{T} \sum_{t=1}^{T} \tau_t^{\max} = \frac{1}{T} \sum_{t=1}^{T} \max_{i \in [N]}\{\tau_t^i\}$, which its bounded-ness is naturally hold if the maximum gradient delay holds.

Assumption 4.3 is common in analyzing asynchronous and anarchic FL algorithms which incorporate the gradient delays into their algorithm design (Koloskova et al., 2022; Yang et al., 2021; Nguyen et al., 2022; Toghani & Uribe, 2022; Wang et al., 2023).

**Assumption 4.4** (Uniform Arrivals of Gradient Computation). Let the set $\mathcal{M}_t$ (with size $M$) include clients that transmit their local updates to the server in global round $t$. We assume that the clients' update arrivals are uniformly distributed, i.e., from a theoretical perspective, the $M$ clients in $\mathcal{M}_t$ are randomly sampled without replacement from all clients $[N]$ according to a uniform distribution.

Assumption 4.4 is also discussed in Anarchic FL (Yang et al., 2022) and FADAS (Wang et al., 2024d). Note that this assumption is only used for the convenience of theoretical analysis. Our experimental settings does not rely on this assumption.

**Theorem 4.5.** *Under Assumptions 4.1-4.4, if the global learning rate satisfies $\eta = \Theta(\sqrt{M})$, the local learning rate satisfies $\eta_l = \Theta(\sqrt{\mathcal{F}}/\sqrt{TK(\sigma_l^2 + K\sigma_g^2)})$ and and distillation learning rate satisfies $\eta_d = \Theta(\mathcal{F}/\sqrt{T^3(\sigma_l^2 + K\sigma_g^2)}Q)$, where $\mathcal{F} = f_1 - f_*$ and $f_* = \arg\min_{\boldsymbol{x}} f(\boldsymbol{x})$, then the global rounds of Algorithm 1 satisfy*

$$\frac{1}{T} \sum_{t=1}^{T} \mathbb{E}[\|\nabla f(\boldsymbol{x}_t)\|^2] = \mathcal{O}\left( \frac{\sqrt{\mathcal{F}}\sigma}{\sqrt{TKM}} + \frac{\sqrt{\mathcal{F}}\sigma_g}{\sqrt{TM}} + \frac{\mathcal{F}\tau_{\max}\tau_{\text{avg}}}{T} + \frac{\sqrt{\mathcal{F}}\nu^2}{T} \right), \tag{5}$$

*Remark* 4.6. Theorem 4.5 indicates that, given a sufficiently large $T$, the proposed FedEcho algorithm achieves a convergence rate of $\mathcal{O}\big(\frac{1}{\sqrt{TM}}\big)$ with respect to both $T$ and $M$.

Compared to existing asynchronous FL methods in non-convex scenarios, the proposed FedEcho achieves a comparable convergence rate and delay dependency, consistent with the analyses in (Wang et al., 2024c) and (Wang et al., 2024d). Moreover, relative to the original analysis of Fed-Buff (Nguyen et al., 2022; Toghani & Uribe, 2022), FedEcho demonstrates a slightly improved

dependency on $\tau_{\max}$ in the convergence rate. It is worth noting that, the bounded stochastic variance $\sigma_d$ for FedEcho contributes solely to a negligible residual term that is asymptotically dominated by $\mathcal{O}\left(\frac{1}{T}\right)$. Moreover, while CA$^2$FL (Wang et al., 2024c) attempts to mitigate the adverse effects of stale updates by incorporating cached model updates $h_t^i$ into global aggregation, its performance highly relies on the practical conditions that these cached updates remain sufficiently informative over time. This limitation is reflected in the theoretical analysis as well, as the convergence rate relies an additional assumption of maximum staleness $\rho_{\max}$, and the rate includes the term $\frac{\zeta_{\max}\sigma_l^2}{T}$. As such, when a client has not participated in the training process for many rounds, a large $\zeta_{\max}$ can significantly degrade the overall convergence rate.

## 5 EXPERIMENTAL RESULTS

We evaluate the performance of our proposed algorithm through experiments on vision and text classification tasks, as well as a natural language generation task. For vision classification, we use CIFAR-10 and CIFAR-100 (Krizhevsky et al., 2009) datasets with ResNet-18 model (He et al., 2016). For text classification, we adopt the MRPC subtask from the GLUE benchmark (Wang et al., 2018) with the BERT-base model (Devlin et al., 2018). To extend our study to natural language generation, we fine-tune the Qwen-2.5 1.5B Instruct model (Team, 2024) on the MathInstruct dataset (Yue et al., 2023). We compare our proposed FedEcho against several representative FL baselines, such as FedBuff (without differential privacy) (Nguyen et al., 2022), CA$^2$FL (Wang et al., 2024c), FedAC (Zang et al., 2024), and FADAS (Wang et al., 2024d). We exclude FedAsync (Xie et al., 2019) from comparison, as its convergence is known to be unstable under significant data heterogeneity and communication delays. The key implementation details are summarized below, and all experiments are conducted on a single NVIDIA A6000 GPU, with additional results and settings provided in Appendix A.

**Overview of running time and delay simulation.** We simulate client runtimes as follows. At the beginning of training, each client is assigned to a runtime category based on a parameter $\gamma$ and the number of local training samples. This design reflects the practical observation that clients with larger datasets sometimes require more training time. In general, a smaller $\gamma$ indicates a stronger dependence of runtime on the number of local samples. To simulate wall-clock runtime, we uniformly sample the runtime of each client from category-specific distributions, as detailed in Table 1. In practice, we ensure that no more than 10% of the clients are assigned to the large-delay group.

Table 1: Details for wall-clock delay simulation (in units of 10 seconds).

| Delay/Runtime | Short | Medium | Long |
|---|---|---|---|
| *Large delay* | $U(1,2)$ | $U(3,5)$ | $U(50,80)$ |
| *Mild delay* | $U(1,2)$ | $U(3,5)$ | $U(10,20)$ |

### 5.1 EXPERIMENTS FOR VISION CLASSIFICATION

**Overview of server-distillation.** For CIFAR-10 experiments, we use CIFAR-100 (Krizhevsky et al., 2009), STL10 (Coates et al., 2011), and synthetic data generated by a denoising diffusion probabilistic model (DDPM)[2] as unlabeled distillation datasets. For CIFAR-100 experiments, we use CIFAR-10 (Krizhevsky et al., 2009), STL10 (Coates et al., 2011), and the same synthetic dataset. The default number of distillation samples is set to 2000, the clipping threshold $\nu$ is fixed at 5, and the minimum and maximum coefficients are set to $\alpha_{\min} = 0.2$ and $\alpha_{\max} = 0.8$, respectively. We use the Adam optimizer for server-side distillation with learning rate $\eta_d = 3 \times 10^{-6}$, $\beta_1 = 0.9$, $\beta_2 = 0.999$, and $\epsilon = 10^{-8}$.

**Overview of data partition and local training details.** We consider the total number of clients of 50, the concurrency $M_c$ is set to 25, and the buffer size $M = 5$. The client data is partitioned using a Dirichlet distribution (Wang et al., 2020a;b), where the concentration parameter $\alpha_{\mathrm{Dir}}$ controls the degree of heterogeneity. For CIFAR-10, we consider two levels of heterogeneity with $\alpha_{\mathrm{Dir}} = 0.1$ and $\alpha_{\mathrm{Dir}} = 0.3$. For CIFAR-100, we adopt $\alpha_{\mathrm{Dir}} = 0.03$ and $\alpha_{\mathrm{Dir}} = 0.1$ due to its larger number of classes. Each client performs two local training epochs per communication round with a mini-batch

---

[2]`https://huggingface.co/google/ddpm-cifar10-32`

Table 2: The test accuracy on the training ResNet-18 model on the CIFAR-10 dataset with two data heterogeneity levels in the mild and large delay scenarios for 500 communication rounds. We report the average accuracy and standard deviation for three different seeds.

| Method | Large delay | | Mild delay | |
|---|---|---|---|---|
| | Dir (0.1) | Dir (0.3) | Dir (0.1) | Dir (0.3) |
| FedBuff | 52.85±7.76 | 51.44±8.74 | 42.57±5.00 | 41.60±5.10 |
| CA$^2$FL | 55.33±7.87 | 50.37±7.27 | 42.99±7.93 | 57.31±7.03 |
| FedAC | 62.06±3.16 | 74.67±1.95 | 69.54±1.43 | 76.23±1.14 |
| FADAS | 61.25±1.16 | 75.43±2.77 | 65.51±2.04 | 74.85±1.30 |
| FedEcho (CIFAR-100) | 75.39±2.93 | 84.11±1.49 | **79.69**±0.68 | 83.74±0.96 |
| FedEcho (STL10) | **77.81**±0.48 | **84.96**±0.86 | 77.99±0.94 | **83.98**±0.67 |
| FedEcho (synthetic w. diffusion) | 74.93±1.75 | 80.68±1.35 | 72.95±5.20 | 81.27±1.71 |

Table 3: The test accuracy on the training ResNet-18 model on the CIFAR-100 dataset with two data heterogeneity levels in the mild and large delay scenarios for 1000 communication rounds.

| Method | Large delay | | Mild delay | |
|---|---|---|---|---|
| | Dir (0.03) | Dir (0.1) | Dir (0.03) | Dir (0.1) |
| FedBuff | 30.58±1.13 | 44.28±0.49 | 32.24±1.70 | 45.41±0.58 |
| CA$^2$FL | 37.32±0.49 | 43.88±0.24 | 35.02±2.25 | 45.03±0.16 |
| FedAC | 50.09±0.64 | 58.07±0.52 | 51.09±0.47 | 58.61±0.29 |
| FADAS | 48.22±0.70 | 57.95±0.40 | 49.29±0.37 | 58.22±0.32 |
| FedEcho (CIFAR-10) | 53.74±0.29 | 61.54±0.14 | **55.55**±0.26 | **63.05**±0.21 |
| FedEcho (STL10) | **53.94**±0.18 | **62.30**±0.09 | 55.26±0.21 | 59.68±0.10 |
| FedEcho (synthetic w. diffusion) | 48.04±0.55 | 57.58±0.13 | 49.23±0.45 | 62.04±0.20 |

size of 50. All baseline methods use SGD with weight decay $10^{-4}$ as the local optimizer, and both global and local learning rates are tuned for each method via grid search.

**Main results.** Table 2 and Table 3 report the test accuracy of ResNet-18 on CIFAR-10 and CIFAR-100 under varying heterogeneity levels and delay scenarios. FedEcho consistently achieves substantially higher accuracy across all settings, regardless of the chosen server-side distillation dataset. Both tables also show that existing asynchronous FL baselines, particularly FedBuff and CA$^2$FL, suffer from pronounced accuracy degradation under asynchronous delays. In contrast, FedEcho demonstrates greater stability, yielding more consistent performance across delay and heterogeneity conditions. On CIFAR-100, FedEcho outperforms the best-performing baseline by at least 3 percentage points. We further observe that external unlabeled datasets such as STL10, as well as cross-dataset CIFAR-100/10, provide more stable and effective distillation targets than synthetic data generated by diffusion models.

**Server computation costs and memory overhead.**

To evaluate the additional computation cost introduced by this design, we analysis the real runtime during server aggregation steps. With the default distillation setting of 2,000 samples, since each distillation step involves the same procedure of back-propagation steps as regular training, the average server-side distillation process in our image classification experiments takes approximately 2.2–3.5 seconds per global round, which is reasonably acceptable.

Moreover, as discussed in previous sections, the proposed FedEcho requires the server to maintain both the global model checkpoints used for each client's local training and the corresponding client logits. To quantify this storage cost, we track

Table 4: The test accuracy on the training BERT-base model on the GLUE-MRPC dataset under large delay scenarios for 200 communication rounds.

| Method | Accuracy |
|---|---|
| FedBuff | 71.47±0.12 |
| CA$^2$FL | 72.14±3.19 |
| FedAC | 73.19±0.85 |
| FADAS | 68.38±0.00 |
| FedEcho | **77.30**±1.31 |

the number of global model checkpoints required. Our experiments show that the server needs to

store at most 8 different checkpoints (i.e., global models from previous rounds), which is acceptable for a central server.

## 5.2 Experiments on Language Tasks

Due to space constraints, we provide the experimental details in Appendix A. As shown in Table 4, FedEcho achieves the highest validation accuracy, demonstrating the effectiveness of uncertainty-aware distillation in exploiting straggler contributions for the text classification task. Furthermore, when applied to generative language models, FedEcho attains the best GSM8K score performance on the Qwen2.5-1.5B Instruct model fine-tuned with MathInstruct (Table 5), improving over the best baseline FedAC and significantly surpassing the original Qwen2.5-1.5B Instruct model. These results confirm that our uncertainty-aware distillation mechanism benefits both language classification and generative reasoning tasks, highlighting its robustness in leveraging straggler contributions.

## 5.3 Ablation Studies

In the following, we analyze several aspects of the proposed FedEcho, including how to choose the number of distillation samples and the impact of mixing weight $\alpha$ in the uncertainty-aware distillation loss. Additionally, we discuss the effect of clipping threshold $\nu$ in Appendix A.

**Number of distillation samples.** We investigate the impact of the number of distillation samples on the server-distillation procedures. Specifically, we consider an experimental setting where we train a ResNet-18 model with a local CIFAR-10 dataset, and use CIFAR-100 as a server distillation set. We compare 500, 1000, 2000, and 5000 distillation samples. As shown in Table 6, increasing the number of distillation samples consistently leads to higher final accuracy.

**Mixing weight $\alpha$ in the uncertainty-aware distillation loss** We investigate the impact of weight $\alpha$ on the uncertainty-aware distillation loss and its effect on overall performance. Specifically, we compare the dynamic $\alpha$ with $\alpha_{\min} = 0.2$, $\alpha_{\max} = 0.8$ with $\alpha = 0$ (CE loss) and $\alpha = 1$ (KL divergence loss) and the mixed loss with fixed $\alpha = 0.5$. Table 7 shows that using only CE loss leads to the lowest accuracy, while applying solely KL divergence or a fixed mixture number improves the performance. Notably, the dynamic $\alpha$ achieves the best overall result, which indicates that balancing CE and KL losses based on the entropy-based uncertainty provides effective guidance for the global model under asynchronous settings.

Table 5: GSM8K accuracy of the Qwen2.5-1.5B Instruct model fine-tuned on the MathInstruct dataset for 5 communication rounds under large-delay scenarios.

| Method | GSM8K acc. |
|---|---|
| Qwen 2.5-1.5B Instruct | 42.38 |
| FedBuff | 52.08 |
| CA$^2$FL | 51.70 |
| FedAC | 52.16 |
| FADAS | 46.40 |
| FedEcho | **52.62** |

## 6 Conclusions

In this work, we addressed the critical challenges of asynchronous FL, where outdated updates from straggler clients and the dominance of faster clients under heterogeneous data distributions often degrade model performance. We proposed FedEcho, a novel uncertainty-aware distillation framework to retain the useful information from the prediction level rather than directly merging stale parameter updates. FedEcho mitigates the dual issues of asynchronous delay and data heterogeneity while preserving the diverse knowledge from all clients. We provide theoretical convergence guarantee, and extensive experiments demonstrate that FedEcho consistently outperforms existing asynchronous FL baselines across both classification and generation tasks, achieving robust and scalable performance without accessing private client data.

Table 6: Ablations of the number of distillation data samples.

| # samples | Acc. |
|---|---|
| 500 | 61.86±6.05 |
| 1000 | 67.91±6.22 |
| 2000 | 75.39±2.93 |
| 5000 | **77.03**±0.43 |

Table 7: Ablations of $\alpha$.

| | Acc. |
|---|---|
| $\alpha = 0$ | 71.01±2.79 |
| $\alpha = 1$ | 74.86±3.14 |
| $\alpha = 0.5$ | 74.92±2.54 |
| dynamic $\alpha$ | **75.39**±2.93 |

## ETHICS STATEMENT

This work focuses on the development and evaluation of federated learning algorithms using publicly available benchmark datasets. No human subjects or personally identifiable information are involved in this study. The research does not raise additional ethical concerns related to privacy, fairness, safety, or potential misuse beyond those already inherent in standard machine learning research.

## REPRODUCIBILITY STATEMENT

We have made detailed efforts to ensure the reproducibility of our results. The experimental setup, including datasets, hyperparameter configurations, and implementation details, is provided in the main text and Appendix A. Complete proofs of the theoretical results and assumptions are included in Appendix C. The datasets we use are publicly available, and we describe all necessary preprocessing steps in the supplementary materials. Code will be released upon acceptance to further facilitate reproducibility.

## STATEMENT OF LLMs USAGE

We use LLMs solely as general-purpose assistive tools for minor tasks such as English proofreading, rephrasing, code debugging, and formatting adjustments.

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

# A ADDITIONAL EXPERIMENTS

## A.1 EXPERIMENTAL SETTINGS

### A.1.1 IMAGE CLASSIFICATION

We report the local and global learning rates for image classification tasks as follows.

Table 8: Learning rates details for CIFAR-10 experiment. The learning rates is reported by $\eta_l/\eta_g$ (local/global).

| Method | Large delay | | Mild delay | |
|---|---|---|---|---|
| | Dir (0.1) | Dir (0.3) | Dir (0.1) | Dir (0.3) |
| FedBuff | 0.01/1.0 | 0.01/1.0 | 0.01/1.0 | 0.01/1.0 |
| CA$^2$FL | 0.01/1.0 | 0.01/1.0 | 0.01/1.0 | 0.01/1.0 |
| FedAC | 0.1/0.0001 | 0.1/0.0001 | 0.1/0.0001 | 0.1/0.0001 |
| FADAS | 0.1/0.0001 | 0.1/0.0001 | 0.1/0.0001 | 0.1/0.0001 |
| FedEcho (CIFAR-100) | 0.03/1.0 | 0.03/1.0 | 0.03/1.0 | 0.03/1.0 |
| FedEcho (STL10) | 0.03/1.0 | 0.03/1.0 | 0.03/1.0 | 0.03/1.0 |
| FedEcho (synthetic w. diffusion) | 0.03/1.0 | 0.03/1.0 | 0.03/1.0 | 0.03/1.0 |

Table 9: Learning rates details for CIFAR-100 experiment. The learning rates is reported by $\eta_l/\eta_g$ (local/global).

| Method | Large delay | | Mild delay | |
|---|---|---|---|---|
| | Dir (0.1) | Dir (0.3) | Dir (0.1) | Dir (0.3) |
| FedBuff | 0.003/1.0 | 0.003/1.0 | 0.003/1.0 | 0.003/1.0 |
| CA$^2$FL | 0.003/1.0 | 0.003/1.0 | 0.003/1.0 | 0.003/1.0 |
| FedAC | 0.1/0.0001 | 0.1/0.0001 | 0.1/0.0001 | 0.1/0.0001 |
| FADAS | 0.1/0.0001 | 0.1/0.0001 | 0.1/0.0001 | 0.1/0.0001 |
| FedEcho (CIFAR-10) | 0.03/1.0 | 0.03/1.0 | 0.03/1.0 | 0.03/1.0 |
| FedEcho (STL10) | 0.03/1.0 | 0.03/1.0 | 0.03/1.0 | 0.03/1.0 |
| FedEcho (synthetic w. diffusion) | 0.03/1.0 | 0.03/1.0 | 0.03/1.0 | 0.03/1.0 |

### A.1.2 TEXT CLASSIFICATION AND LANGUAGE GENERATION

**Overview of experimental setups.** For the text classification tasks on MRPC, we use the validation dataset as evaluation set, and the unlabeled test as the distillation set. For the natural language generation, we randomly select 100 mathematical queries and exclude them from the training set. The default number of distillation samples is set to 100. Same as the previous image classification task, we set $\nu = 5$, $\alpha_{\min} = 0.2$ and $\alpha_{\max} = 0.8$. We use the Adam optimizer for server-side distillation with learning rate $\eta_d = 3 \times 10^{-7}$ for classification experiment and $\eta_d = 3 \times 10^{-6}$ for generation, $\beta_1 = 0.9$, $\beta_2 = 0.999$, and $\epsilon = 10^{-8}$. We consider the total number of clients of 10, the concurrency $M_c$ is set to 5, and the buffer size $M = 3$. The client data is partitioned using a Dirichlet distribution with $\alpha_{\mathrm{Dir}} = 0.6$ Each client performs one local training epochs per communication round with a mini-batch size of 8. We adopt AdamW as the local optimizer, and the local and global learning rate list are summarized in Table 10.

## A.2 ADDITIONAL DISCUSSIONS

As shown in Eq. (2), we incorporate gradient clipping into the distillation loss. This is particularly important in the early stages of training, when teacher logits may be inaccurate and fail to fully capture local knowledge. To mitigate such imperfect guidance, we constrain the magnitude of the distillation gradient using a clipping threshold. In this section, we conduct an ablation study to examine the impact of different clipping thresholds on the overall performance. As shown in Table 11,

Table 10: Learning rates details for CIFAR-100 experiment. The learning rates is reported by $\eta_l/\eta_g$ (local/global).

| Method | MRPC | MathInstruct |
|--------|------|--------------|
| FedBuff | 3e-5/1.0 | 3e-5/1.0 |
| CA$^2$FL | 3e-5/1.0 | 3e-5/1.0 |
| FedAC | 3e-6/0.1 | 3e-6/0.1 |
| FADAS | 3e-6/0.1 | 3e-6/0.1 |
| FedEcho | 3e-5/1.0 | 3e-5/1.0 |

setting $\nu = 5$ yields the best accuracy, while smaller ($\nu = 1$) or no clipping ($\nu = \infty$) slightly reduces performance, highlighting the importance of a moderate threshold.

Table 11: Ablations of clipping threshold $\nu$.

|  | Acc. |
|--------|------|
| $\nu = 1$ | 74.38±2.67 |
| $\nu = 5$ | 75.39±2.93 |
| $\nu = \infty$ | 74.14±2.07 |

# B   DISCUSSION OF DISTILLATION GRADIENT

**Classification with a single hidden layer.**   Given the original loss function with input data $a_n$ and ground truth $b_n$ on the server, the server aims to optimize $\min f(\boldsymbol{x}) = \min \ell(\phi_{\boldsymbol{x}}(a_n), b_n)$. In this section, we investigate the distillation process defined as $\min \ell(\phi_{\boldsymbol{x}}(a_n), \frac{1}{N}\sum_{i=1}^{N}(\phi_{\boldsymbol{x}_i}(a_n)) := \min f_d(\boldsymbol{x}|\boldsymbol{x}_1, ..., \boldsymbol{x}_N)$ which is applied at the end of each global round. For notational simplicity, we omit the global round index $t$ and refer to $\boldsymbol{x}$ as the current global model and $\boldsymbol{x}_i$ are local models.

Consider a $K$-classification model with one hidden layer, i.e., $\phi_{\boldsymbol{x}}(a_n) = \text{Softmax}(\boldsymbol{x}_T a_n) \in \mathbb{R}^K$, where $\boldsymbol{x} \in \mathbb{R}^{d \times K}$ are the student model parameters, suppose there is an input data $a_n \in \mathcal{A} = \mathbb{R}^d$. Then we have the following related to distillation loss

$$f(\boldsymbol{x}|\boldsymbol{x}_1, ..., \boldsymbol{x}_n) = \ell(\phi_{\boldsymbol{x}}(a_n), \frac{1}{N}\sum_{i=1}^{N}(\phi_{\boldsymbol{x}_i}(a_n)))$$

$$= \ell(\sigma(\boldsymbol{x}_T a_n), s_n)$$

$$= -\sum_{k=1}^{K} s_{n,k} \log \sigma(\boldsymbol{x}_T a_n)_k$$

$$= -\sum_{k=1}^{K} s_{n,k} \log \frac{e^{(\boldsymbol{x}_1)^T a_n}}{\sum_{j=1}^{K} e^{(\boldsymbol{x}_j)^T a_n}}$$

$$= \sum_{k=1}^{K} s_{n,k}(\log \sum_{j=1}^{K} e^{(\boldsymbol{x}_j)^T a_n} - \log e^{(\boldsymbol{x}_k)^T a_n})$$

$$= \sum_{k=1}^{K} s_{n,k} \log \sum_{j=1}^{K} e^{(\boldsymbol{x}_j)^T a_n} - \sum_{k=1}^{K} s_{n,k} \log e^{(\boldsymbol{x}_k)^T a_n}$$

$$= \log \sum_{k=1}^{K} e^{(\boldsymbol{x}_k)^T a_n} - \sum_{k=1}^{K} s_{n,k} \log e^{(\boldsymbol{x}_k)^T a_n}$$

$$= \log \sum_{k=1}^{K} e^{(\boldsymbol{x}_k)^T a_n} - \sum_{k=1}^{K} s_{n,k}(\boldsymbol{x}_k)^T a_n, \tag{6}$$

where the second to last one is due to the sum of soft labels are equal to 1. Therefore,

$$\nabla_{\boldsymbol{x}_k} f(\boldsymbol{x}|\boldsymbol{x}_1, ..., \boldsymbol{x}_n) = \nabla_{\boldsymbol{x}_k}[\log \sum_{k=1}^{K} e^{(\boldsymbol{x}_k)^T a_n} - \sum_{k=1}^{K} s_{n,k}(\boldsymbol{x}_k)^T a_n]$$

$$= \frac{e^{(\boldsymbol{x}_k)^T a_n}}{\sum_{i=1}^{K} e^{(\boldsymbol{x}_i)^T a_n}} a_n - s_{n,k} a_n$$

$$= (\sigma(\boldsymbol{x}_T a_n) - s_n)_k a_n$$

$$= (\sigma(\boldsymbol{x}_T a_n) - b_n)_k a_n - (\frac{1}{N}\sum_{i=1}^{N} \sigma(\boldsymbol{x}_i^T a_n) - b_n)_k a_n$$

$$= \nabla_{\boldsymbol{x}_k} f(\boldsymbol{x}) - \frac{1}{N}\sum_{i=1}^{N} \nabla_{\boldsymbol{x}_i^k} f(\boldsymbol{x}_i). \tag{7}$$

Therefore, we can conclude the distillation gradient as

$$\nabla_{\boldsymbol{x}} f(\boldsymbol{x}|\boldsymbol{x}_1, ..., \boldsymbol{x}_n) = \nabla_{\boldsymbol{x}} f(\boldsymbol{x}) - \frac{1}{N}\sum_{i=1}^{N} \nabla_{\boldsymbol{x}_i} f(\boldsymbol{x}_i). \tag{8}$$

**Generic classification.** Consider an arbitrary neural network architecture for classification that ends with a softmax layer. Let $\psi(\boldsymbol{x})$ denote the logits produced by the model with parameters $\boldsymbol{x}$. Define the loss function associated with logits $z$ and true label $b_n$ as $\phi_{\boldsymbol{x}}(z) = \ell(\text{Softmax}(z), b_n)$, where $\ell$ is a loss function. In this setting, $\psi(\boldsymbol{x})$ computes the logits from the input data, and $\phi$ evaluates the loss given the logits. We define the overall loss function as $f(\boldsymbol{x}) = \phi(\psi(\boldsymbol{x}))$. Then, the following relationship holds:

$$a_n \to \psi_n(\boldsymbol{x}) \to \phi_{\boldsymbol{x}}(a_n) := \text{Softmax}(\psi_n(\boldsymbol{x})) \in \mathbb{R}^K. \tag{9}$$

Then we have the following for the distillation loss

$$f(\boldsymbol{x}|\boldsymbol{x}_1,...,\boldsymbol{x}_n) = \ell(\phi_{\boldsymbol{x}}(a_n), \frac{1}{N}\sum_{i=1}^{N}(\phi_{\boldsymbol{x}_i}(a_n)))$$

$$= \ell(\sigma(\psi_n(\boldsymbol{x})), \frac{1}{N}\sum_{i=1}^{N}\sigma(\psi_n(\boldsymbol{x}_i)))$$

$$= \ell(\sigma(\psi_n(\boldsymbol{x})), s_n)$$

$$= -\sum_{k=1}^{K} s_{n,k} \log \sigma(\psi_n(\boldsymbol{x}))_k$$

$$= -\sum_{k=1}^{K} s_{n,k} \log \frac{e^{\psi_{n,k}(\boldsymbol{x})}}{\sum_{j=1}^{K} e^{\psi_{n,j}(\boldsymbol{x})}}$$

$$= \sum_{k=1}^{K} s_{n,k} (\log \sum_{j=1}^{K} e^{\psi_{n,j}(\boldsymbol{x})} - \psi_{n,k}(\boldsymbol{x}))$$

$$= \sum_{k=1}^{K} s_{n,k} \log \sum_{j=1}^{K} e^{\psi_{n,j}(\boldsymbol{x})} - \sum_{k=1}^{K} s_{n,k}\psi_{n,k}(\boldsymbol{x})$$

$$= \log \sum_{k=1}^{K} e^{\psi_{n,k}(\boldsymbol{x})} - \sum_{k=1}^{K} s_{n,k}\psi_{n,k}(\boldsymbol{x}). \tag{10}$$

Applying the gradient operator, we have

$$\nabla_{\boldsymbol{x}} f(\boldsymbol{x}|\boldsymbol{x}_1,...,\boldsymbol{x}_n) = \nabla_{\boldsymbol{x}}[\log \sum_{k=1}^{K} e^{\psi_{n,k}(\boldsymbol{x})} - \sum_{k=1}^{K} s_{n,k}\psi_{n,k}(\boldsymbol{x})]$$

$$= \sum_{k=1}^{K} \frac{e^{\psi_{n,k}(\boldsymbol{x})}}{\sum_{j=1}^{K} e^{\psi_{n,j}(\boldsymbol{x})}} \nabla_{\boldsymbol{x}}\psi_{n,k}(\boldsymbol{x}) - \sum_{k=1}^{K} s_{n,k} \nabla_{\boldsymbol{x}}\psi_{n,k}(\boldsymbol{x})$$

$$= \sum_{k=1}^{K} (\sigma(\psi_n(\boldsymbol{x})) - s_n)_k \nabla_{\boldsymbol{x}}\psi_{n,k}(\boldsymbol{x})$$

$$= \frac{\partial \psi_n(\boldsymbol{x})}{\partial \boldsymbol{x}}(\sigma(\psi_n(\boldsymbol{x})) - s_n). \tag{11}$$

Therefore, we have

$$\sigma(\psi_n(\boldsymbol{x})) - s_n = (\sigma(\psi_n(\boldsymbol{x})) - b_n) - (\frac{1}{N}\sum_{i=1}^{N}\sigma(\psi_n(\boldsymbol{x}_i)) - b_n)$$

$$= \nabla_\psi \ell(\sigma(\psi_n(\boldsymbol{x})), b_n) - \frac{1}{N}\sum_{i=1}^{N}\nabla_\psi \ell(\sigma(\psi_n(\boldsymbol{x}_i)), b_n)$$

$$= \nabla\varphi_n(\psi_n(\boldsymbol{x})) - \frac{1}{N}\sum_{i=1}^{N}\nabla\varphi_n(\psi_n(\boldsymbol{x}_i)), \tag{12}$$

further if taking into account that $f_n(\boldsymbol{x}) = \varphi_n(\psi_n(\boldsymbol{x}))$, we show the following for the distillation gradient,

$$\nabla_{\boldsymbol{x}} f(\boldsymbol{x}|\boldsymbol{x}_1,...,\boldsymbol{x}_n) = \frac{\partial \psi_n(\boldsymbol{x})}{\partial \boldsymbol{x}}(\nabla\varphi_n(\psi_n(\boldsymbol{x})) - \frac{1}{N}\sum_{i=1}^{N}\nabla\varphi_n(\psi_n(\boldsymbol{x}_i)))$$

$$= \frac{\partial \psi_n(\boldsymbol{x})}{\partial \boldsymbol{x}}\frac{\partial f_n(\boldsymbol{x})}{\partial \psi_n(\boldsymbol{x})} - \frac{1}{N}\sum_{i=1}^{N}\frac{\partial \psi_n(\boldsymbol{x})}{\partial \boldsymbol{x}}\frac{\partial f_n(\boldsymbol{x}_i)}{\partial \psi_n(\boldsymbol{x}_i)}. \tag{13}$$

The first term is the student' gradient, while the second term differs from the average of the teachers' gradient $\frac{1}{N}\sum_{i=1}^{N}\nabla_{\boldsymbol{x}_i} f(\boldsymbol{x}_i)$ as the partial derivatives of logits are with respect to the student model.

## C  CONVERGENCE ANALYSIS

*Proof of Theorem 4.5.* Following several previous works studied asynchronous federated methods (Chen et al., 2018; Wang et al., 2022), we have

From Assumption 4.1, $f$ is $L$-smooth, taking the total expectation over all previous round, $0, 1, ..., t-1$ on the auxiliary sequence $\boldsymbol{x}_t$,

$$
\begin{aligned}
&\mathbb{E}[f(\boldsymbol{x}_{t+1}) - f(\boldsymbol{x}_t)] \\
&= \mathbb{E}[f(\boldsymbol{x}_{t+1}) - f(\widehat{\boldsymbol{x}}_{t+1}) + f(\widehat{\boldsymbol{x}}_{t+1}) - f(\boldsymbol{x}_t)] \\
&= \underbrace{\mathbb{E}[f(\boldsymbol{x}_{t+1}) - f(\widehat{\boldsymbol{x}}_{t+1})]}_{E_{\text{distill}}} + \underbrace{\mathbb{E}[f(\widehat{\boldsymbol{x}}_{t+1}) - f(\boldsymbol{x}_t)]}_{E_{\text{original}}},
\end{aligned} \tag{14}
$$

where by Assumption 4.1, there is

$$
\begin{aligned}
\mathbb{E}[f(\boldsymbol{x}_{t+1}) - f(\widehat{\boldsymbol{x}}_{t+1})] &\leq \mathbb{E}[\langle \nabla f(\widehat{\boldsymbol{x}}_{t+1}), \boldsymbol{x}_{t+1} - \widehat{\boldsymbol{x}}_{t+1}\rangle] + \frac{L}{2}\mathbb{E}[\|\boldsymbol{x}_{t+1} - \widehat{\boldsymbol{x}}_{t+1}\|^2] \\
&= \underbrace{\mathbb{E}[\langle \nabla f(\widehat{\boldsymbol{x}}_{t+1}), \eta_d \widehat{\boldsymbol{\Delta}}_t\rangle]}_{I_1} + \underbrace{\frac{L}{2}\mathbb{E}[\|\widehat{\boldsymbol{\Delta}}_t\|^2]}_{I_2},
\end{aligned} \tag{15}
$$

and

$$
\begin{aligned}
\mathbb{E}[f(\widehat{\boldsymbol{x}}_{t+1}) - f(\boldsymbol{x}_t)] &\leq \mathbb{E}[\langle \nabla f(\boldsymbol{x}_t), \widehat{\boldsymbol{x}}_{t+1} - \boldsymbol{x}_t\rangle] + \frac{L}{2}\mathbb{E}[\|\widehat{\boldsymbol{x}}_{t+1} - \boldsymbol{x}_t\|^2] \\
&= \underbrace{\mathbb{E}[\langle \nabla f(\boldsymbol{x}_t), \eta\boldsymbol{\Delta}_t\rangle]}_{I_3} + \underbrace{\frac{\eta^2 L}{2}\mathbb{E}[\|\boldsymbol{\Delta}_t\|^2]}_{I_4}.
\end{aligned} \tag{16}
$$

**Bounding $I_1$** For $I_1$, there is

$$
\begin{aligned}
I_1 &= \eta_d \mathbb{E}[\langle \nabla f(\widehat{\boldsymbol{x}}_{t+1}), \widehat{\boldsymbol{\Delta}}_t\rangle] \\
&= \eta_d \mathbb{E}[\langle \nabla f(\widehat{\boldsymbol{x}}_{t+1}), \widehat{\boldsymbol{\Delta}}_t + Q\nabla f(\widehat{\boldsymbol{x}}_{t+1}) - Q\nabla f(\widehat{\boldsymbol{x}}_{t+1})\rangle] \\
&= -\eta_d Q\mathbb{E}[\|\nabla f(\widehat{\boldsymbol{x}}_{t+1})\|^2] + \eta_d \mathbb{E}[\langle \nabla f(\widehat{\boldsymbol{x}}_{t+1}), \widehat{\boldsymbol{\Delta}}_t + Q\nabla f(\widehat{\boldsymbol{x}}_{t+1})\rangle] \\
&= -\eta_d Q\mathbb{E}[\|\nabla f(\widehat{\boldsymbol{x}}_{t+1})\|^2] + \eta_d \mathbb{E}[\langle \nabla f(\widehat{\boldsymbol{x}}_{t+1}), -\sum_{q=1}^{Q}\nabla f_d(\widehat{\boldsymbol{x}}_{t+1,q}; \xi) + Q\nabla f(\widehat{\boldsymbol{x}}_{t+1})\rangle],
\end{aligned} \tag{17}
$$

by the fact of $\langle \boldsymbol{a}, \boldsymbol{b}\rangle = \frac{1}{2}[\|\boldsymbol{a}\|^2 + \|\boldsymbol{b}\|^2 - \|\boldsymbol{a} - \boldsymbol{b}\|^2]$, for second term in (24), we have

$$
\begin{aligned}
&\eta_d \mathbb{E}[\langle \nabla f(\widehat{\boldsymbol{x}}_{t+1}), -\sum_{q=1}^{Q}\nabla f_d(\widehat{\boldsymbol{x}}_{t+1,q}; \xi) + Q\nabla f(\widehat{\boldsymbol{x}}_{t+1})\rangle] \\
&= \eta_d \mathbb{E}[\langle \sqrt{Q}\nabla f(\widehat{\boldsymbol{x}}_{t+1}), -\frac{\sqrt{Q}}{Q}\sum_{q=1}^{Q}[\nabla f_d(\widehat{\boldsymbol{x}}_{t+1,q}) - \nabla f(\widehat{\boldsymbol{x}}_{t+1})]\rangle] \\
&= \frac{\eta_d Q}{2}\mathbb{E}[\|\nabla f(\widehat{\boldsymbol{x}}_{t+1})\|^2] + \frac{\eta_d}{2Q}\mathbb{E}[\|\sum_{q=1}^{Q}[\nabla f_d(\widehat{\boldsymbol{x}}_{t+1,q}) - \nabla f(\widehat{\boldsymbol{x}}_{t+1})]\|^2] \\
&\quad - \frac{\eta_d}{2Q}\mathbb{E}[\|\sum_{q=1}^{Q}\nabla f_d(\widehat{\boldsymbol{x}}_{t+1,q})\|^2],
\end{aligned} \tag{18}
$$

then for the second term, we have

$$\frac{\eta_d}{2Q}\mathbb{E}[\|\sum_{q=1}^{Q}[\nabla f_d(\widehat{\boldsymbol{x}}_{t+1,q}) - \nabla f(\widehat{\boldsymbol{x}}_{t+1})]\|^2]$$

$$\leq \frac{\eta_d}{2}\sum_{q=1}^{Q}\mathbb{E}[\|\nabla f_d(\widehat{\boldsymbol{x}}_{t+1,q}) - \nabla f(\widehat{\boldsymbol{x}}_{t+1})\|^2]$$

$$\leq \eta_d\sum_{q=1}^{Q}\mathbb{E}[\|\nabla f_d(\widehat{\boldsymbol{x}}_{t+1,q})\|^2] + \eta_d\sum_{q=1}^{Q}\mathbb{E}[\|\nabla f(\widehat{\boldsymbol{x}}_{t+1})\|^2]$$

$$\leq \eta_d Q\nu^2 + \eta_d Q\mathbb{E}[\|\nabla f(\widehat{\boldsymbol{x}}_{t+1})\|^2]. \tag{19}$$

Thus we have

$$I_1 \leq -\frac{\eta_d Q}{2}\mathbb{E}[\|\nabla f(\widehat{\boldsymbol{x}}_{t+1})\|^2] + \eta_d Q\nu^2 + \eta_d Q\mathbb{E}[\|\nabla f(\widehat{\boldsymbol{x}}_{t+1})\|^2]$$

$$\leq \frac{\eta_d Q}{2}\mathbb{E}[\|\nabla f(\widehat{\boldsymbol{x}}_{t+1})\|^2] + \eta_d Q\nu^2. \tag{20}$$

For the first term in the previous inequality, there is

$$\mathbb{E}[\|\nabla f(\widehat{\boldsymbol{x}}_{t+1})\|^2] = \mathbb{E}[\|\nabla f(\widehat{\boldsymbol{x}}_{t+1}) - \nabla f(\boldsymbol{x}_t) + \nabla f(\boldsymbol{x}_t)\|^2]$$

$$\leq 2L^2\mathbb{E}[\|\widehat{\boldsymbol{x}}_{t+1} - \boldsymbol{x}_t\|^2] + 2\mathbb{E}[\|\nabla f(\boldsymbol{x}_t)\|^2]$$

$$= 2\eta^2 L^2\mathbb{E}[\|\boldsymbol{\Delta}_t\|^2] + 2\mathbb{E}[\|\nabla f(\boldsymbol{x}_t)\|^2]. \tag{21}$$

By Lemma D.2, there is

$$\frac{1}{N}\sum_{i=1}^{N}\mathbb{E}[\|\boldsymbol{x}_{t-\rho_t^i,K} - \boldsymbol{x}_{t-\rho_t^i}\|^2] \leq 5K\eta_l^2(\sigma_l^2 + 6K\sigma_g^2) + 30K^2\eta_l^2\frac{1}{N}\sum_{i=1}^{N}\mathbb{E}[\|\nabla f(\boldsymbol{x}_{t-\rho_t^i})\|^2]. \tag{22}$$

thus

$$I_1 + I_2 \leq \eta^2\eta_d QL^2\mathbb{E}[\|\boldsymbol{\Delta}_t\|^2] + \eta_d Q\mathbb{E}[\|\nabla f(\boldsymbol{x}_t)\|^2] + \eta_d Q\nu^2 + \frac{L}{2}\mathbb{E}[\|\widehat{\boldsymbol{\Delta}}_t\|^2]. \tag{23}$$

**Bounding** $I_3$ Denote a sequence $\bar{\boldsymbol{\Delta}}_t = -\frac{\eta_l}{N}\sum_{i\in[N]}\sum_{k=0}^{K-1}\boldsymbol{g}_{t-\tau_t^i,k}^i = -\frac{\eta_l}{N}\sum_{i\in[N]}\sum_{k=0}^{K-1}\nabla F_i(\boldsymbol{x}_{t-\tau_t^i,k}^i;\xi)$, where $\xi \sim \mathcal{D}_i$. For $I_3$, there is

$$I_3 = \eta\mathbb{E}[\langle\nabla f(\boldsymbol{x}_t), \boldsymbol{\Delta}_t\rangle]$$

$$= \eta\mathbb{E}[\langle\nabla f(\boldsymbol{x}_t), \bar{\boldsymbol{\Delta}}_t\rangle]$$

$$= \eta\mathbb{E}[\langle\nabla f(\boldsymbol{x}_t), \bar{\boldsymbol{\Delta}}_t + \eta_l K\nabla f(\boldsymbol{x}_t) - \eta_l K\nabla f(\boldsymbol{x}_t)\rangle]$$

$$= -\eta\eta_l K\mathbb{E}[\|\nabla f(\boldsymbol{x}_t)\|^2] + \eta\mathbb{E}[\langle\nabla f(\boldsymbol{x}_t), \bar{\boldsymbol{\Delta}}_t + \eta_l K\nabla f(\boldsymbol{x}_t)\rangle]$$

$$= -\eta\eta_l K\mathbb{E}[\|\nabla f(\boldsymbol{x}_t)\|^2] + \eta\mathbb{E}[\langle\nabla f(\boldsymbol{x}_t), -\frac{\eta_l}{N}\sum_{i\in[N]}\sum_{k=0}^{K-1}\nabla F_i(\boldsymbol{x}_{t-\tau_t^i,k}^i;\xi_i) + \frac{\eta_l K}{N}\sum_{i\in[N]}\nabla F_i(\boldsymbol{x}_t)\rangle], \tag{24}$$

where the second equality holds due to the characteristic of uniform arrivals (see Assumption 4.3), thus $\mathbb{E}(\boldsymbol{\Delta}_t) = \bar{\boldsymbol{\Delta}}_t$. The last inequality holds by the definition of $\bar{\boldsymbol{\Delta}}_t$ and the fact of the objective function $f(\boldsymbol{x}) = \frac{1}{N}\sum_{i=1}^{N}F_i(\boldsymbol{x})$. By the fact of $\langle\boldsymbol{a}, \boldsymbol{b}\rangle = \frac{1}{2}[\|\boldsymbol{a}\|^2 + \|\boldsymbol{b}\|^2 - \|\boldsymbol{a} - \boldsymbol{b}\|^2]$, for second

term in (24), we have

$$
\eta \mathbb{E}[\langle \nabla f(\boldsymbol{x}_t), -\frac{\eta_l}{N} \sum_{i \in [N]} \sum_{k=0}^{K-1} \boldsymbol{g}_{t-\tau_t^i, k}^i + \frac{\eta_l K}{N} \sum_{i \in [N]} \nabla F_i(\boldsymbol{x}_t) \rangle]
$$

$$
= \eta \mathbb{E}[\langle \sqrt{\eta_l K} \nabla f(\boldsymbol{x}_t), -\sqrt{\eta_l K} \frac{1}{NK} \sum_{i \in [N]} \sum_{k=0}^{K-1} (\boldsymbol{g}_{t-\tau_t^i, k}^i - \nabla F_i(\boldsymbol{x}_t)) \rangle]
$$

$$
= \eta \mathbb{E}[\langle \sqrt{\eta_l K} \nabla f(\boldsymbol{x}_t), -\sqrt{\eta_l K} \frac{1}{NK} \sum_{i \in [N]} \sum_{k=0}^{K-1} (\nabla F_i(\boldsymbol{x}_{t-\tau_t^i, k}^i) - \nabla F_i(\boldsymbol{x}_t)) \rangle]
$$

$$
= \frac{\eta \eta_l K}{2} \mathbb{E}[\|\nabla f(\boldsymbol{x}_t)\|^2] + \frac{\eta \eta_l}{2N^2 K} \mathbb{E}[\|\sum_{i \in [N]} \sum_{k=0}^{K-1} (\nabla F_i(\boldsymbol{x}_{t-\tau_t^i, k}^i) - \nabla F_i(\boldsymbol{x}_t))\|^2]
$$

$$
- \frac{\eta \eta_l}{2N^2 K} \mathbb{E}[\sum_{i \in [N]} \sum_{k=0}^{K-1} \nabla F_i(\boldsymbol{x}_{t-\tau_t^i, k}^i)\|^2], \tag{25}
$$

where the second equality holds by $\mathbb{E}[\boldsymbol{g}_{t-\tau_t^i, k}^i] = \mathbb{E}[\nabla F_i(\boldsymbol{x}_{t-\tau_t^i, k}^i)]$. Then for the second term in Eq. (25) , we have

$$
\frac{\eta \eta_l}{2N^2 K} \mathbb{E}[\|\sum_{i \in [N]} \sum_{k=0}^{K-1} (\nabla F_i(\boldsymbol{x}_{t-\tau_t^i, k}^i) - \nabla F_i(\boldsymbol{x}_t))\|^2]
$$

$$
\leq \frac{\eta \eta_l}{2N^2 K} \mathbb{E}[\|\sum_{i \in [N]} \sum_{k=0}^{K-1} (\nabla F_i(\boldsymbol{x}_{t-\tau_t^i, k}^i) - \nabla F_i(\boldsymbol{x}_t))\|^2]
$$

$$
\leq \frac{\eta \eta_l}{2N} \sum_{i \in [N]} \sum_{k=0}^{K-1} \mathbb{E}[\|\nabla F_i(\boldsymbol{x}_t) - \nabla F_i(\boldsymbol{x}_{t-\tau_t^i, k}^i)\|^2]
$$

$$
\leq \frac{\eta \eta_l}{N} \sum_{i \in [N]} \sum_{k=0}^{K-1} [\mathbb{E}[\|\nabla F_i(\boldsymbol{x}_t) - \nabla F_i(\boldsymbol{x}_{t-\tau_t^i})\|^2] + \mathbb{E}[\|\nabla F_i(\boldsymbol{x}_{t-\tau_t^i}) - \nabla F_i(\boldsymbol{x}_{t-\tau_t^i, k}^i)\|^2]]
$$

$$
\leq \frac{\eta \eta_l}{N} \sum_{i \in [N]} \sum_{k=0}^{K-1} [L^2 \mathbb{E}[\|\boldsymbol{x}_t - \boldsymbol{x}_{t-\tau_t^i}\|^2] + L^2 \mathbb{E}[\|\boldsymbol{x}_{t-\tau_t^i} - \boldsymbol{x}_{t-\tau_t^i, k}^i\|^2]], \tag{26}
$$

where the second inequality holds by $\forall \boldsymbol{a}_i, \|\sum_{i=1}^n \boldsymbol{a}_i\|^2 \leq n \sum_{i=1}^n \|\boldsymbol{a}_i\|^2$, and the last inequality holds by Assumption 4.1. For the second term in Eq. (26), following by Lemma D.2, there is

$$
\mathbb{E}[\|\boldsymbol{x}_{t-\tau_t^i} - \boldsymbol{x}_{t-\tau_t^i, k}^i\|^2] = \mathbb{E}[\|\sum_{m=0}^{k-1} \eta_l \boldsymbol{g}_{t-\tau_t^i, m}^i\|^2]
$$

$$
\leq 5K\eta_l^2 (\sigma_l^2 + 6K\sigma_g^2) + 30K^2 \eta_l^2 \mathbb{E}[\|\nabla f(\boldsymbol{x}_{t-\tau_t^i})\|^2]. \tag{27}
$$

For the first term in Eq. (26), since by $\forall \boldsymbol{a}_i, \|\sum_{i=1}^n \boldsymbol{a}_i\|^2 \leq n \sum_{i=1}^n \|\boldsymbol{a}_i\|^2$, there is

$$
\mathbb{E}[\|\boldsymbol{x}_t - \boldsymbol{x}_{t-\tau_t^i}\|^2] = \mathbb{E}[\|\sum_{s=t-\tau_t^i}^{t-1} (\boldsymbol{x}_{s+1} - \widehat{\boldsymbol{x}}_{s+1} + \widehat{\boldsymbol{x}}_{s+1} - \boldsymbol{x}_s)\|^2]
$$

$$
\leq 2\tau_t^i \sum_{s=t-\tau_t^i}^{t-1} \mathbb{E}[\|\boldsymbol{x}_{s+1} - \widehat{\boldsymbol{x}}_{s+1}\|^2] + 2\tau_t^i \sum_{s=t-\tau_t^i}^{t-1} \mathbb{E}[\|\widehat{\boldsymbol{x}}_{s+1} - \boldsymbol{x}_s\|^2]
$$

$$
\leq 2\tau_t^i \sum_{s=t-\tau_t^i}^{t-1} \mathbb{E}[\|\eta \boldsymbol{\Delta}_s\|^2] + 2\tau_t^i \sum_{s=t-\tau_t^i}^{t-1} \mathbb{E}[\|\widehat{\boldsymbol{\Delta}}_s\|^2], \tag{28}
$$

Plugging Eq. (25), Eq. (26) and Eq. (27) to (24), we have

$$
\begin{aligned}
\mathbb{E}[I_3] \leq &- \frac{\eta\eta_l K}{2}\mathbb{E}[\|\nabla f(\boldsymbol{x}_t)\|^2] - \frac{\eta\eta_l}{2K}\mathbb{E}[\|\frac{1}{N}\sum_{i=1}^{N}\sum_{k=0}^{K-1}\nabla F_i(\boldsymbol{x}_{t-\tau_t^i,k}^i)\|^2] \\
&+ \eta\eta_l KL^2[5K\eta_l^2(\sigma_l^2 + 6K\sigma_g^2) + 30K^2\eta_l^2\frac{1}{N}\sum_{i=1}^{N}\mathbb{E}[\|\nabla f(\boldsymbol{x}_{t-\tau_t^i})\|^2]] \\
&+ \frac{2\eta\eta_l}{N}\sum_{i\in[N]}\sum_{k=0}^{K-1}L^2\tau_t^i[\mathbb{E}[\|\widehat{\boldsymbol{\Delta}}_s\|^2] + \mathbb{E}[\|\eta\boldsymbol{\Delta}_s\|^2]].
\end{aligned} \tag{29}
$$

**Merging pieces.** Therefore, by merging pieces together, we have

$$
\begin{aligned}
\mathbb{E}[f(\boldsymbol{x}_{t+1}) - f(\boldsymbol{x}_t)] &= \mathbb{E}[I_1 + I_2 + I_3 + I_4] \\
\leq &- \frac{\eta\eta_l K}{2}\mathbb{E}[\|\nabla f(\boldsymbol{x}_t)\|^2] - \frac{\eta\eta_l}{2K}\mathbb{E}[\|\frac{1}{N}\sum_{i=1}^{N}\sum_{k=0}^{K-1}\nabla F_i(\boldsymbol{x}_{t-\tau_t^i,k}^i)\|^2] \\
&+ \eta\eta_l KL^2[5K\eta_l^2(\sigma_l^2 + 6K\sigma_g^2) + 30K^2\eta_l^2\frac{1}{N}\sum_{i=1}^{N}\mathbb{E}[\|\nabla f(\boldsymbol{x}_{t-\tau_t^i})\|^2]] \\
&+ \frac{2\eta\eta_l}{N}\sum_{i\in[N]}\sum_{k=0}^{K-1}L^2\tau_t^i\sum_{s=t-\tau_t^i}^{t-1}[\mathbb{E}[\|\eta\boldsymbol{\Delta}_s\|^2] + \mathbb{E}[\|\widehat{\boldsymbol{\Delta}}_s\|^2]] + \frac{\eta^2 L}{2}\mathbb{E}[\|\boldsymbol{\Delta}_t\|^2] \\
&+ \eta^2\eta_d QL^2\mathbb{E}[\|\boldsymbol{\Delta}_t\|^2] + \eta_d Q\mathbb{E}[\|\nabla f(\boldsymbol{x}_t)\|^2] + \eta_d Q\nu^2 + \frac{\eta_d^2 L}{2}\mathbb{E}[\|\widehat{\boldsymbol{\Delta}}_t\|^2] \\
\leq &- \frac{\eta\eta_l K}{2}\mathbb{E}[\|\nabla f(\boldsymbol{x}_t)\|^2] - \frac{\eta\eta_l}{2K}\mathbb{E}[\|\frac{1}{N}\sum_{i=1}^{N}\sum_{k=0}^{K-1}\nabla F_i(\boldsymbol{x}_{t-\tau_t^i,k}^i)\|^2] \\
&+ \eta\eta_l KL^2[5K\eta_l^2(\sigma_l^2 + 6K\sigma_g^2) + 30K^2\eta_l^2\frac{1}{N}\sum_{i=1}^{N}\mathbb{E}[\|\nabla f(\boldsymbol{x}_{t-\tau_t^i})\|^2]] \\
&+ 2\eta\eta_l KL^2\tau_t^i\sum_{s=t-\tau_t^i}^{t-1}[\mathbb{E}[\|\eta\boldsymbol{\Delta}_s\|^2] + \mathbb{E}[\|\widehat{\boldsymbol{\Delta}}_s\|^2]] + \frac{\eta^2 L}{2}\mathbb{E}[\|\boldsymbol{\Delta}_t\|^2] \\
&+ \eta^2\eta_d QL^2\mathbb{E}[\|\boldsymbol{\Delta}_t\|^2] + \eta_d Q\mathbb{E}[\|\nabla f(\boldsymbol{x}_t)\|^2] + \eta_d Q\nu^2 + \frac{\eta_d^2 L}{2}\mathbb{E}[\|\widehat{\boldsymbol{\Delta}}_t\|^2].
\end{aligned} \tag{30}
$$

Thus, summing up, we have

$$\sum_{t=1}^{T} \mathbb{E}[f(\boldsymbol{x}_{t+1}) - f(\boldsymbol{x}_t)]$$

$$\leq - \frac{\eta \eta_l K}{2} \sum_{t=1}^{T} \mathbb{E}[\|\nabla f(\boldsymbol{x}_t)\|^2] - \frac{\eta \eta_l}{2K} \sum_{t=1}^{T} \mathbb{E}[\|\frac{1}{N} \sum_{i=1}^{N} \sum_{k=0}^{K-1} \nabla F_i(\boldsymbol{x}_{t-\tau_t^i, k}^i)\|^2]$$

$$+ T \eta \eta_l K L^2 [5K \eta_l^2 (\sigma_l^2 + 6K \sigma_g^2) + 30K^2 \eta_l^2 \frac{1}{N} \sum_{i=1}^{N} \sum_{t=1}^{T} \mathbb{E}[\|\nabla f(\boldsymbol{x}_{t-\tau_t^i})\|^2]]$$

$$+ 2\eta \eta_l K L^2 \frac{1}{N} \sum_{i=1}^{N} \sum_{t=1}^{T} \tau_t^i \sum_{s=t-\tau_t^i}^{t-1} [\mathbb{E}[\eta \boldsymbol{\Delta}_s\|^2] + \mathbb{E}[\eta_d \widehat{\boldsymbol{\Delta}}_s\|^2]] + \frac{\eta^2 L}{2} \sum_{t=1}^{T} \mathbb{E}[\|\boldsymbol{\Delta}_t\|^2]$$

$$+ \eta^2 \eta_d Q L^2 \sum_{t=1}^{T} \mathbb{E}[\|\boldsymbol{\Delta}_t\|^2] + \eta_d Q \sum_{t=1}^{T} \mathbb{E}[\|\nabla f(\boldsymbol{x}_t)\|^2] + \eta_d T Q \nu^2 + \frac{\eta_d^2 L}{2} \sum_{t=1}^{T} \mathbb{E}[\|\widehat{\boldsymbol{\Delta}}_t\|^2]$$

$$\leq - \frac{\eta \eta_l K}{2} \sum_{t=1}^{T} \mathbb{E}[\|\nabla f(\boldsymbol{x}_t)\|^2] - \frac{\eta \eta_l}{2K} \sum_{t=1}^{T} \mathbb{E}[\|\frac{1}{N} \sum_{i=1}^{N} \sum_{k=0}^{K-1} \nabla F_i(\boldsymbol{x}_{t-\tau_t^i, k}^i)\|^2]$$

$$+ T \eta \eta_l K L^2 [5K \eta_l^2 (\sigma_l^2 + 6K \sigma_g^2) + 30K^2 \eta_l^2 \tau_{\max} \frac{1}{N} \sum_{i=1}^{N} \sum_{t=1}^{T} \mathbb{E}[\|\nabla f(\boldsymbol{x}_t)\|^2]]$$

$$+ 2\eta \eta_l K L^2 \tau_{\max} \tau_{avg} \sum_{t=1}^{T} [\mathbb{E}[\|\eta \boldsymbol{\Delta}_t\|^2] + \mathbb{E}[\|\eta_d \widehat{\boldsymbol{\Delta}}_t\|^2]] + \frac{\eta^2 L}{2} \sum_{t=1}^{T} \mathbb{E}[\|\boldsymbol{\Delta}_t\|^2]$$

$$+ \eta^2 \eta_d Q L^2 \sum_{t=1}^{T} \mathbb{E}[\|\boldsymbol{\Delta}_t\|^2] + \eta_d Q \sum_{t=1}^{T} \mathbb{E}[\|\nabla f(\boldsymbol{x}_t)\|^2] + \eta_d T Q \nu^2 + \frac{\eta_d^2 L}{2} \sum_{t=1}^{T} \mathbb{E}[\|\widehat{\boldsymbol{\Delta}}_t\|^2], \quad (31)$$

By applying Lemma D.1 to the $E[|\boldsymbol{\Delta}_t|^2]$ term and Lemma D.3 to the $E[|\widehat{\boldsymbol{\Delta}}_t|^2]$ term, and under appropriate conditions on the learning rate, we obtain

$$\sum_{t=1}^{T} \mathbb{E}[f(\boldsymbol{x}_{t+1}) - f(\boldsymbol{x}_t)] \leq - \frac{\eta \eta_l K}{2} \sum_{t=1}^{T} \mathbb{E}[\|\nabla f(\boldsymbol{x}_t)\|^2]$$

$$+ T \eta \eta_l K L^2 [5K \eta_l^2 (\sigma_l^2 + 6K \sigma_g^2)]$$

$$+ \left(2\eta \eta_l K L^2 \tau_{\max} \tau_{avg} + \frac{\eta^2 L}{2} + \eta^2 \eta_d Q L^2\right) T \left(\frac{\eta_l^2 K}{M} \sigma_l^2 + \frac{\eta_l^2 K^2}{M} \sigma_g^2\right)$$

$$+ T \eta_d Q \nu^2 + \left(\frac{\eta_d^2 L}{2} + 2\eta \eta_l \eta_d^2 K L^2 \tau_{\max} \tau_{avg}\right) T (5Q(\sigma_d^2 + 2Q\nu^2))$$

$$\Rightarrow$$

$$\frac{1}{T} \sum_{t=1}^{T} \mathbb{E}[\|\nabla f(\boldsymbol{x}_t)\|^2] \leq \frac{2}{T \eta \eta_l K} [f_1 - f_*] + 2L^2 [5K \eta_l^2 (\sigma_l^2 + 6K \sigma_g^2)]$$

$$+ \left(4\eta_l K L^2 \tau_{\max} \tau_{avg} + \eta L + 2\eta^2 \eta_d Q L^2\right) \left(\frac{\eta_l K}{M} \sigma_l^2 + \frac{\eta_l K}{M} \sigma_g^2\right)$$

$$+ \frac{2\eta_d Q \nu^2}{\eta \eta_l K} + \left(\frac{\eta_d^2 L}{\eta \eta_l K} + 4\eta_d^2 L^2 \tau_{\max} \tau_{avg}\right) (5Q(\sigma_d^2 + 2Q\nu^2)), \quad (32)$$

If the global learning rate satisfies $\eta = \Theta(\sqrt{M})$, the local learning rate satisfies $\eta_l = \Theta(\sqrt{\mathcal{F}}/\sqrt{TK(\sigma_l^2 + K\sigma_g^2)})$ and and distillation learning rate satisfies $\eta_d = \Theta(\mathcal{F}/\sqrt{T^3(\sigma_l^2 + K\sigma_g^2)Q})$, where $\mathcal{F} = f_1 - f_*$ and $f_* = \operatorname{argmin}_{\boldsymbol{x}} f(\boldsymbol{x})$, then the global rounds of

Algorithm 1 satisfy

$$\frac{1}{T}\sum_{t=1}^{T}\mathbb{E}[\|\nabla f(\boldsymbol{x}_t)\|^2] = \mathcal{O}\left(\frac{\sqrt{\mathcal{F}}\sigma}{\sqrt{TKM}} + \frac{\sqrt{\mathcal{F}}\sigma_g}{\sqrt{TM}} + \frac{\mathcal{F}\tau_{\max}\tau_{\mathrm{avg}}}{T} + \frac{\sqrt{\mathcal{F}}\nu^2}{T}\right). \tag{33}$$

$\square$

# D    SUPPORTING LEMMAS

**Lemma D.1.** *Recall the sequence* $\boldsymbol{\Delta}_t = \frac{1}{M}\sum_{i\in\mathcal{M}_t}\boldsymbol{\Delta}_{t-\tau_t^i}^i = -\frac{\eta_l}{M}\sum_{i\in\mathcal{M}_t}\sum_{k=0}^{K-1}\boldsymbol{g}_{t-\tau_t^i,k}^i = -\frac{\eta_l}{M}\sum_{i\in\mathcal{M}_t}\sum_{k=0}^{K-1}\nabla F_i(\boldsymbol{x}_{t-\tau_t^i,k}^i;\xi)$ *and* $\mathcal{M}_t$ *be the set that include client send the local updates to the server at global round* $t$. *The global model difference* $\boldsymbol{\Delta}_t$ *satisfies*

$$\mathbb{E}[\|\boldsymbol{\Delta}_t\|^2] = \mathbb{E}\left[\left\|\frac{1}{M}\sum_{i\in\mathcal{M}_t}\boldsymbol{\Delta}_{t-\tau_t^i}^i\right\|^2\right]$$

$$\leq \frac{2K\eta_l^2}{M}\sigma_l^2 + \frac{2\eta_l^2(N-M)}{NM(N-1)}\left[15NK^3L^2\eta_l^2(\sigma_l^2 + 6K\sigma_g^2) + (90NK^4L^2\eta_l^2 + 3K^2)\right.$$

$$\left.\cdot\sum_{i=1}^{N}\mathbb{E}[\|\nabla f(\boldsymbol{x}_{t-\tau_t^i})\|^2] + 3NK^2\sigma_g^2\right] + \frac{2\eta_l^2(M-1)}{NM(N-1)}\mathbb{E}\left[\left\|\sum_{i=1}^{N}\sum_{k=0}^{K-1}\nabla F_i(\boldsymbol{x}_{t-\tau_t^i,k}^i)\right\|^2\right].$$

*Proof.* The proof of Lemma D.1 is similar to the proof of Lemma C.6 in (Wang et al., 2022). $\square$

**Lemma D.2.** *(This lemma follows from Lemma 3 in FedAdam (Reddi et al., 2021). For local learning rate which satisfying* $\eta_l \leq \frac{1}{8KL}$, *the local model difference after* $k$ *($\forall k \in \{0, 1, ..., K-1\}$) steps local updates satisfies*

$$\frac{1}{N}\sum_{i=1}^{N}\mathbb{E}[\|\boldsymbol{x}_{t,k}^i - \boldsymbol{x}_t\|^2] \leq 5K\eta_l^2(\sigma_l^2 + 6K\sigma_g^2) + 30K^2\eta_l^2\mathbb{E}[\|\nabla f(\boldsymbol{x}_t)\|^2]. \tag{34}$$

*Proof.* The proof of Lemma D.2 is similar to the proof of Lemma 3 in Reddi et al. (2021). $\square$

**Lemma D.3.** *For the intermediate update for distillation, we have*

$$\mathbb{E}[\|\widehat{\boldsymbol{x}}_{t+1,q} - \widehat{\boldsymbol{x}}_{t+1}\|^2] \leq 5\eta_d^2 Q(\sigma_d^2 + 2Q\nu^2). \tag{35}$$

*Proof.*

$$\mathbb{E}[\|\widehat{\boldsymbol{x}}_{t+1,q} - \widehat{\boldsymbol{x}}_{t+1}\|^2]$$

$$= \mathbb{E}[\|\widehat{\boldsymbol{x}}_{t+1,q-1} - \widehat{\boldsymbol{x}}_{t+1} - \eta_d\nabla f_d(\widehat{\boldsymbol{x}}_{t+1,q-1};\xi)\|^2]$$

$$= \mathbb{E}[\|\widehat{\boldsymbol{x}}_{t+1,q-1} - \widehat{\boldsymbol{x}}_{t+1} - \eta_d\nabla f_d(\widehat{\boldsymbol{x}}_{t+1,q-1};\xi) + \eta_d\nabla f_d(\widehat{\boldsymbol{x}}_{t+1,q-1}) - \eta_d\nabla f_d(\widehat{\boldsymbol{x}}_{t+1,q-1})\|^2]$$

$$= (1 + \frac{1}{2Q-1})\mathbb{E}[\|\widehat{\boldsymbol{x}}_{t+1,q-1} - \widehat{\boldsymbol{x}}_{t+1}\|^2] + \eta_d^2\sigma_d^2 + 2\eta_d^2 Q\mathbb{E}[\|\nabla f_d(\widehat{\boldsymbol{x}}_{t+1,q-1})\|^2]$$

$$\leq (1 + \frac{1}{2Q-1})\mathbb{E}[\|\widehat{\boldsymbol{x}}_{t+1,q-1} - \widehat{\boldsymbol{x}}_{t+1}\|^2] + \eta_d^2\sigma_d^2 + 2\eta_d^2 Q\nu^2, \tag{36}$$

given the fact that $(1 + \frac{1}{2Q-1})^Q \leq 5$ for $Q > 1$, there is

$$\mathbb{E}[\|\widehat{\boldsymbol{x}}_{t+1,q} - \widehat{\boldsymbol{x}}_{t+1}\|^2] \leq 5\eta_d^2 Q(\sigma_d^2 + 2Q\nu^2). \tag{37}$$

$\square$

