# OpenReview forum: "Stragglers Can Contribute More: Uncertainty-Aware Distillation for Asynchronous Federated Learning"
_ICLR.cc/2026/Conference — ICLR 2026 Conference Withdrawn Submission_

### Official Review · Reviewer_5fSL · 2025-10-28

**Soundness:** 3
**Presentation:** 4
**Contribution:** 3
**Rating:** 6
**Confidence:** 3

**Summary:**

The paper targets the issues raised by stragglers in AFL. The authors propose FedEcho to aggregate logits and use uncertainty-aware distillation to exploit straggler updates without contaminating server weights. FedEcho aims at mitigating the negative impacts of outdated updates and data heterogeneity. Experiments results show consistent performance advantages under varying delay and heterogeneity setups. The authors also provide a theoretical convergence analysis of FedEcho proving a convergence rate of O( 1/√TM).

**Strengths:**

The paper is well written with clear motivation and description of the proposal.

The proposed entropy-based weight adaptation between KL and CE is reasonable and interesting. Though there are entropy-based works in the field, one of which is also mentioned in Related works (Itahara et al 2021), the proposal is still somewhat novel.

The authors have provided both thoeretical analysis and emprical tests to show the performance boundries.

**Weaknesses:**

The choice of alpha min and max values lack of discussion.
The intuition behind entropy and alpha is reasonable, but lack deeper and wider discussion and exploration.
The limitation of the work is not discussed.

**Questions:**

1. Can the authors elaborate on how the min and max values of alpha are determined and why?

2. The logic of entropy-based weight assignment is reasonable, however the intuition "that when teacher predictions are highly uncertain, the hard labels derived from them may be unreliable, and relying more on soft labels helps the global student model avoid learning from inaccurate or noisy signals" needs more clarification and discussion.
A. Are there other reasons than "unreliable" when high entropy occurs?
B. High entropy can also indicate that neither hard labels nor soft labels are beneficial. Could the authors elaborate on this with theoretical analysis or discussion? I

---

### Official Review · Reviewer_3zpx · 2025-10-30

**Soundness:** 3
**Presentation:** 2
**Contribution:** 2
**Rating:** 2
**Confidence:** 3

**Summary:**

The paper proposes FedEcho, a novel asynchronous federated learning (FL) framework to solve the conflict between mitigating stale updates from slow "straggler" clients and preventing bias from fast clients. Instead of directly averaging stale parameters, FedEcho uses server-side knowledge distillation. It reconstructs each client's model to generate predictions (logits) on a public dataset, then refines the global model by distilling knowledge from an ensemble of these client predictions. Experiments show FedEcho consistently outperforms state-of-the-art baselines.

**Strengths:**

1. The use of uncertainty-aware distillation to extract knowledge from stragglers without direct parameter mixing is an elegant and effective solution to the core staleness-vs-bias problem in asynchronous FL.
2. The method is convincingly validated against strong baselines across diverse tasks, including vision, NLP, and generative language models, demonstrating robust and significant performance gains.

**Weaknesses:**

1. Unclear Algorithmic Rationale and Complexity: (This addresses your second question). The algorithm presented is confusing because it seems to perform two separate update steps. In Line 11, the global model is updated via standard parameter averaging (x_bt+1 = xt + η∆t), which directly incorporates stale updates—the very problem the paper aims to avoid. Then, in a second phase (Lines 12-16), this newly updated model is further refined via distillation. The paper does not adequately justify why this two-step process is necessary or optimal. The motivation suggests distillation is an alternative to parameter averaging for stragglers, but the algorithm implements it as an add-on to parameter averaging for everyone. This dual-update mechanism feels unnecessarily complex and somewhat contradicts the initial motivation.
2. The paper fails to cite or compare against highly relevant recent work, notably "Momentum-Driven Adaptivity: Towards Tuning-Free Asynchronous Federated Learning". That work also explicitly claims to address the joint challenges of large asynchronous delays and data heterogeneity, and importantly, it aims for a parameter-free design. FedEcho, in contrast, introduces a significant number of new hyperparameters (ηd, Q, ν, αmin, αmax, the size and choice of unlabeled dataset U).

[1] Yan, W., Zhong, X., Wang, X., & Zhang, Y. J. A. Momentum-Driven Adaptivity: Towards Tuning-Free Asynchronous Federated Learning. In Forty-second International Conference on Machine Learning, (2025).

3. Significant Practical Overhead: The paper somewhat downplays the practical costs of the server-side operations. The server must:

* Store up to Mc historical global model checkpoints.
* Store logits for all N clients on the unlabeled dataset U.
* Perform computationally expensive inference for each arriving client update to generate new logits.
* Run Q backpropagation steps for distillation after every M client updates.

This represents a substantial increase in computation and memory overhead compared to standard asynchronous methods. Calling this "reasonably acceptable" may not hold for very large models or federated networks with many clients.


4. Reliance on a Public Unlabeled Dataset: The method's effectiveness is contingent on the availability of a suitable public dataset U. While the authors show this can be synthetic or out-of-domain, it is still a significant logistical requirement that many other FL algorithms do not have, potentially limiting its real-world applicability.

**Questions:**

1. What's the definition of $p_c^u$ in Eq. (4)?
2. Why use distillation rather than directly handling asynchronous delays?

---

### Official Review · Reviewer_HHHK · 2025-10-31

**Soundness:** 2
**Presentation:** 2
**Contribution:** 2
**Rating:** 2
**Confidence:** 4

**Summary:**

This work aims to achieve asynchronous federated learning without the degradation of the model performance due to updates from the straggler clients with stale model updates and the bias from faster clients, under data heterogeneity across clients. To achieve this goal, the paper proposes FedEcho that utilizes uncertainty aware distillation for the global model updates from the local updates of clients. The method prioritizes the predictions with higher confidence for distillation to prevent model degradation. The paper provides convergence analysis that aligns with previous asynchronous FL convergence bounds and includes experiments that show FedEcho outperforms other methods.

**Strengths:**

- The paper tackles a relevant problem in federated learning where there are a number of stragglers and often waiting for them can be a communication bottleneck, and their stale updates can hurt the global model performance.
- The paper includes convergence guarantees for the method to show that the method's theoretical convergence aligns with previous work in asynchronous FL.
- The paper is clearly written and easy to understand, with additional ablation studies of FedEcho on the different hyperparameters such as the number of distillation data samples or the mixing weight $\alpha$.

**Weaknesses:**

- The novelty is rather limited in the sense that the main contribution here is using the loss function in (3) for distillation with the mixing weight $\alpha$ that balances between hard labels and soft ones. Such approach has already been proposed in many other works [1-2], although not directly in the context of asynchronous FL. Moreover, with just (3), it seems like a weak argument to claim that uncertainty is leveraged here, especially when tuning $\alpha$ can seem to be a bit tricky here.
- I also have couple of doubts in the effectiveness of FedEcho, especially because it seems that the distribution of the public dataset as well as the value of $\alpha$ will matter a lot in its success and these can be hard to control since the server would not have direct access to the client's data and not really know its distribution. Moreover, since the server has to pass all the clients' local updates and generate logits to perform distillation, in cases like next work prediction for LLM decoders, the logit space can be at least 10K in vocab size and would be impractical to use.
- I also have concerns regarding the experiment settings where the total number of clients is only limited to 50. Stragglers usually become more relevant and a problem when there are more number of clients with only a small number of selected clients. The settings that the authors use here does not really seem to reflect the realistic scenarios where asynchronous FL may actually be relevant.

[1] Heterogeneous Ensemble Knowledge Transfer for Training Large Models in Federated Learning (IJCAI 2022)
[2] DFRD: Data-Free Robustness Distillation for Heterogeneous Federated Learning (Neurips 2022)

**Questions:**

Please address the questions.

---

### Note · Authors · 2026-01-24

I have read and agree with the venue's withdrawal policy on behalf of myself and my co-authors.